# Catalyst Design: Counter Anion Effect on Ni Nanocatalysts Anchored on Hollow Carbon Spheres

**DOI:** 10.3390/nano13030426

**Published:** 2023-01-20

**Authors:** Ryan O’Connor, Joyce B. Matsoso, Victor Mashindi, Pumza Mente, Lebohang Macheli, Beatriz D. Moreno, Bryan P. Doyle, Neil J. Coville, Dean H. Barrett

**Affiliations:** 1DSI-NRF Centre of Excellence in Strong Materials, University of the Witwatersrand, WITS, Johannesburg 2050, South Africa; 2Molecular Science Institute, School of Chemistry, University of the Witwatersrand, WITS, Johannesburg 2050, South Africa; 3Department of Inorganic Chemistry, University of Chemistry and Technology in Prague, Dejvice 6, 166 28 Prague 6, Czech Republic; 4Institute of Physical Chemistry, Polish Academy of Science, 01-224 Warsaw, Poland; 5Department of Physics, University of Johannesburg, P.O. Box 524, Auckland Park 2006, South Africa; 6Canadian Light Source Inc., 44 Innovation Boulevard, Saskatoon, SK S7N 2V3, Canada

**Keywords:** catalyst design, counter anion, acetate, chloride, nickel nanoparticles, hollow carbon spheres

## Abstract

Herein, the influence of the counter anion on the structural properties of hollow carbon spheres (HCS) support was investigated by varying the nickel metal precursor salts applied. TEM and SEM micrographs revealed the dimensional dependence of the HCS shell on the Ni precursor salt, as evidenced by thick (~42 nm) and thin (~23 nm) shells for the acetate and chloride-based salts, respectively. Importantly, the effect of the precursor salt on the textural properties of the HCS nanosupports (~565 m^2^/g_Ni(acet)_) and ~607 m^2^/g_NiCl_), influenced the growth of the Ni nanoparticles, viz for the acetate-(*ca* 6.4 nm)- and chloride (*ca* 12 nm)-based salts, respectively. Further, XRD and PDF analysis showed the dependence of the reduction mechanism relating to nickel and the interaction of the nickel–carbon support on the type of counter anion used. Despite the well-known significance of the counter anion on the size and crystallinity of Ni nanoparticles, little is known about the influence of such counter anions on the physicochemical properties of the carbon support. Through this study, we highlight the importance of the choice of the Ni-salt on the size of Ni in Ni–carbon-based nanocatalysts.

## 1. Introduction

The discovery of fullerenes by Kroto and co-workers [1] has widely been considered the birthplace of interest in the study of modern day carbon-based nanomaterials. Today it is largely recognised that carbon has the most identified allotropes of all elements with the likes of carbon black, diamond, fullerenes, graphene, and carbon nanotubes comprising some of the most well-known. Many of these allotropes are known to possess useful properties such as high surface areas, thermal stability, chemical stability under harsh conditions, and excellent electronic conductivity [1,2,3]. Owing to such attractive properties, many carbon allotropes have been explored as popular candidates in electrocatalysis as metal catalyst supports [4,5] as their physicochemical and/or electronic properties facilitate good dispersion of metal nanoparticles, resulting in improved catalytic performance and stability.

Among the explored carbon-based materials, hollow carbon spheres (HCSs), the sister allotrope to fullerenes, have gained significant attention for application as a metal catalyst support due to their unique properties such as encapsulation ability, controllable permeability, surface functionality, high surface-to-volume ratios, and excellent chemical and thermal stabilities [4,6,7]. Typically, the HCSs are classified as hollow-structured carbon capsules comprising a thin shell housing with a void of millimeter, micro or nanometer size [8,9]. For HCSs to be considered as candidates for the application as nanoreactors, two main factors must be taken into consideration: (i) the HCSs synthesis procedure, and (ii) the supported metal nanoparticle synthesis and its interaction with the HCSs. Regarding the synthesis procedure, numerous methods have been reported, often making use of hard or soft template-based strategies [4,9,10]. During the hard-templating route, readily prepared rigid spheres, of desired size and with a narrow particle size distribution, such as silica, zeolites or polymers are employed as the “hard” core templates. They are later sacrificed via etching with acid, base or thermal treatment after the carbon shell has been formed [7,11,12]. On the other hand, via the soft-templating route, the direct generation of HCSs is achieved through the self-assembly of colloidal systems, such as emulsion droplets, micelles, vesicles, or gas bubbles, originating from carbonaceous precursor molecules, surfactants or additives [13,14]. The soft-templating route is attractive due to the ease of removal of the soft core, however, the morphology, structural properties and mono dispersity of the as-prepared HCSs is more difficult to control. Therefore, the hard-templating route is more commonly used for the HCSs synthesis, despite the less-environmentally friendly process required to remove some templates such as silica or metallic spheres [8,15].

The hard-templating route allows for easy tailoring of various properties of the as-produced HCSs such as the inner and outer shell diameters, shell thickness, surface properties, and textural properties. This has led researchers to extensively explore HCSs as excellent candidates for confining metal nanoparticles within the shell of the HCSs to improve their electrocatalytic performance [7,16,17,18]. Encapsulation of metal nanoparticles has been shown to prevent their agglomeration, sintering, leaching as well as their deactivation via reactions with by-products or poisoning of the nanoparticles. Encapsulation also provides control of nanoparticle size thus providing enhanced transport of reactants and by-products. These factors influence the catalytic activity, selectivity, long-term stability and durability of catalysts [19,20].

Although the HCSs-encapsulated metal nanoparticles have been used in a variety of electrocatalytic, environmental, biological, and optoelectronic applications [7,16,17,18], the choice of the carbon precursor plays a significant role during the synthesis of HCSs. For instance, Mezzavila et al. showed that by using a surfactant-assisted simultaneous polycondensation of silica and resorcinol polymer precursors in a colloidal suspension, the hard-templating synthesis procedure for HCSs could be greatly simplified [21]. Their results revealed the formation of hollow mesoporous carbon spheres with a 2–20 nm shell porosity, high surface area (916–2385 m^2^·g^−1^), and a thick shell thickness (60–110 nm). Inspired by the classic Stöber method, Fuertes et al. prepared thin-shelled HCSs (20–30 nm) by enveloping a silica core using a surfactant-free thin layer of resorcinol-formaldehyde (RF) resin [22]. In both cases, the requirement of the use of hydrofluoric acid (HF) to etch away the silica core constituted safety issues when making hollow carbon spheres. To solve the problem of using HF, researchers have used polymer spheres, such as polystyrene (PS) spheres, as a soft template that can be thermally decomposed and removed by heating (~400 °C) [12,23,24]. More importantly, monodispersed PS spheres can be readily prepared, thus enabling easy control of the size of the resultant HCSs, whilst thin carbon shells (<10 nm) can be tailored by adjusting the mass ratio of the carbon precursor to that of the PS spheres. Moreover, the PS spheres, can be functionalized to improve electrocatalytic activity, mesoporosity, and metal nanoparticle anchoring capacity via enhanced polymer–metal interactions [12,25,26].

It is well known that metals have been added to HCSs both inside and outside the HCS shell. This is typically achieved by the incipient wetness impregnation method [27,28,29]. Usually, metal nitrates of nickel, iron and cobalt are often used as the precursor salts, although they often lead to the production of poorly dispersed metal/metal oxide nanoparticles. Other metal precursor salts have also been explored to manipulate the metal–support interactions and hence the catalyst particle size [30,31,32]. In spite of well-documented evidence that the metal anion can affect the size/dispersion of the active phase on a carbon support, what is rarely considered is the effect the anions, in the precursor salt, have on the support itself. In this work, the properties of the Ni encapsulated in HCS nanoreactors (Ni@HCSs) were investigated by studying the influence of the counter anion of the nickel precursors on the resulting morphologies and physicochemical properties of the as-synthesized HCSs encapsulates.

## 2. Materials and Methods

### 2.1. Synthesis of Functionalized Polystyrene Template

Polyacrylic acid (PAA) functionalized polystyrene spheres (PSS) were used as the template and synthesized based on a procedure described elsewhere [12]. Typically, 0.1 g polyvinylpyrrolidone (PVP, 99%, Sigma Aldrich, Johannesburg, South Africa) was dissolved in a mixture of 100 mL deionized water, and 25 mL ethanol (EtOH, 99.9%, Sigma). Thereafter, 8 mL of styrene (C_6_H_5_CH=CH_2_, 99.9%, Merck, Johannesburg, South Africa) and 0.65 mL acrylic acid (CH_2_=CHCOOH, 99.9%, Sigma-Merck) were added to the solution, followed by a 15 min room temperature stirring. Finally, ~2 mM solution of potassium persulfate (K_2_S_2_O_8_, 99.99%, Merck) was added to the above mixture. This was stirred for a further 30 min and heated to 80 °C with continuous stirring for 24 h. To obtain the final templates the materials were cleaned with multiple volumes of ethanol through centrifugation at 18,000 rpm for 15 min at 10 °C and finally dried overnight at 60 °C.

### 2.2. Synthesis of Ni@HCSs Nanocatalysts

Using the as-prepared functionalized PSS template, ~10wt.% Ni was loaded onto the template [12]. To achieve this, ~2.0 g of the functionalized template was dispersed in two separate 5 mL flasks each containing a mixture of 18 mL ethanol and 50 mL deionized water. Ni nanoparticles were then anchored on the templates by stirring ~360 mg of either nickel (II) acetate tetrahydrate (Ni(acetate)_2_, (Ni(OCOCH_3_)_2_.4H_2_O, 99.995%, Merck, Johannesburg, South Africa) or nickel (II) chloride hexahydrate (NiCl_2_.6H_2_O, 98%, Merck, Johannesburg, South Africa) precursor salts with each template mixture. Following the reduction of Ni^2+^ to Ni^0^ by dropwise addition of 27 mL of 8% hydrazine hydrate solution (78–82%, NH_2_NH_2_.H_2_O, Merck, Johannesburg, South Africa) to each reaction vessel, the solutions were stirred at room temperature for 20 h, generating a solution with a pale blue final colour. The mixtures were vacuum filtered, washed with multiple volumes of water and dried in the oven at 60 °C for 12 h, to produce Ni/PSS products. To synthesize the Ni@HCSs nanocatalysts, ~1.8 g of the Ni/PSS_Ni(acet)2_ was dispersed in a solution of 75 mL of absolute ethanol (C_2_H_5_OH, ≥99.5%, Merck) and 40 mL of deionized water. To this solution, a dispersion of 1.0 g of cetyltrimethylammonium bromide (CTAB, (CH_3_(CH_2_)_15_N(Br)(CH_3_)_3_, ≥98%, Sigma Aldrich, Johannesburg, South Africa), 0.8 g of crushed resorcinol (C_6_H_4_-1,3-(OH)_2_, ≥99%, Merck, Johannesburg, South Africa), 0.8 mL of formaldehyde (CH_2_O, 37%, Sigma Aldrich, Johannesburg, South Africa) were added, followed by addition of 10 mL of ammonia solution. Similarly, ~1.8 g of the Ni/PSS_NiCl2_ sample was mixed with 40 mL absolute ethanol and 30 mL of deionized water, then 0.8 g of crushed resorcinol and 0.8 mL of formaldehyde, 2 g of CTAB, and 10 mL of ammonia solution were added. After continuous stirring for 20 h at room temperature, the powdered samples of the Ni/PSS composites were obtained through vacuum filtration, washed with distilled water and ethanol, and dried in the oven at 60 °C for 12 h. The dried products were loaded into quartz boats, placed inside a quartz tube and then placed at the thermocenter of a horizontal chemical vapour deposition (CVD) furnace. The furnace was heated to 350 °C at a rate of 10 °C/min under 50 sccm of nitrogen gas (N_2_, 99.999%, Afrox SA) and held isothermally for 1 h to decompose the polymer spheres. Further heating to 600 °C for 2 h under Ar atmosphere facilitated pyrolysis of the RF layer around the Ni nanoparticles. After growth, the furnace was cooled to room temperature under a continuous flow of N_2_ and the collected samples were labelled as Ni@HCSs_Ni(acet)_ and Ni@HCSs_NiCl_ for samples prepared from Ni(acet)_2_ and NiCl_2_ precursor salts, respectively. The choice of the anions in the salt precursor was governed by the erosion of the carbon by chloride, whereas the acetate groups are known to increase the thickness of the HCSs walls. As a control sample, the pristine HCSs samples were prepared similarly as performed for the nanocatalysts, but without the addition of the metal salt precursor [12,33].

### 2.3. Characterization

The morphological and structural properties of the Ni@ HCSs nanocatalysts were investigated utilizing various characterization techniques. The thermal stability properties of the samples were monitored by thermal gravimetric analysis (TGA) with weight loss derivative outputs (DTG) using a Perkin Elmer 6000 thermogravimetric analyser. Transmission electron microscopy (TEM) images were collected on a Tecnai Spirit T12 at 120 kV acceleration voltage to determine the low magnification morphology of the as-grown carbon nanostructures supported on Cu grids. Zeiss Sigma 300VP scanning electron microscopy (SEM), at an accelerating voltage of 80 kV, was used to determine the morphology of the nanomaterials, after the samples were dispersed in ethanol, dropped on an aluminium stub and coated with a layer (10 nm) of gold-palladium. The phase identification of the samples was confirmed using powder X-ray diffraction (PXRD) with data was collected using a Bruker D2 Phaser diffractometer with a Cu anode producing X-rays at 1.54 Å. The metal particle sizes were determined using Eva diffraction software by applying the Scherrer equation to the most prominent peaks from the corresponding PXRD diffractograms. The degree of crystallization of the samples was determined in the backscattering geometry at 514.5 nm laser excitation wavelength using the Jobin-Yvon T64000 Ultraviolet (UV) LabRam HR-micro-Raman spectrometer with an Olympus BX41 microscope and a liquid nitrogen-cooled charge-coupled device detector. Total scattering data were collected on the Canadian Light Source (CLS) Brockhouse high-energy wiggler beamline using a wavelength of λ = 0.2081 Å and a Perkin Elmer XRD1621 area detector placed 160 mm after the sample. The data were processed using GSAS-II. The Qmax used to produce the pair distribution function (PDF) of the measured samples was 16 Å^−1^. X-ray photoemission spectroscopy (XPS) spectra of the samples were obtained with an overall energy resolution of 0.6 eV using a SPECS PHOIBOS 150 hemispherical electron energy analyser and a monochromatized Al K_α_ photon source (hν = 1486.71 eV). With the use of a low-energy flood gun (electron energy = 2 eV, current = 20 μA), charging of the sample surface was prevented. The textural properties of the as-prepared samples were acquired from the Micrometrics Tristar 3000 surface area and porosity analyser, after degassing the samples for 4 h at 150 °C in N_2_ atmosphere; followed by the acquisition of the adsorption and desorption isotherms of ultra-pure N_2_ gas.

## 3. Results and Discussion

### 3.1. Morphological Analysis

Morphological analysis of the template, pristine HCSs and Ni@HCSs was performed using scanning electron microscopy (SEM) and transmission electron microscopy (TEM). Appendix A revealed that the PSS template consisted of monodispersed nanospheres of ~331 ± 24 nm. SEM micrographs of the pristine HCSs (Figure 1a) depict the formation of completely spherical structures of ~372 ± 28 nm in diameter. On the other hand, a few defective spheres were also evident as seen by the presence of ‘cup-like’ structures and is (i) an indication of incomplete coverage of the polystyrene template spheres (PSS) with the resorcinol formaldehyde or (ii) breakage of the HCSs during the work up procedures. SEM data for the Ni@HCS materials showed that after the loading of Ni nanoparticles onto the PSS template, coverage with RF, and finally carbonization with RF, retention of the spherical architecture of the nanospheres was maintained. However, the use of the NiCl_2_ precursor salt (Figure 1b) led to increased structural deformation, as shown by the presence of holes in the shells of the HCSs (~347 ± 35 nm) as well as the formation of broken shells. On the other hand, the use of the carbon-containing nickel precursor salt (Ni(acet)_2_), led to the formation of more densely interconnected spheres (~332 ± 27 nm), exhibiting minimal structural damage (Figure 1c). The dense interconnectivity suggests the formation of a thicker carbon shell, which could be attributed to the presence of defects and/or dangling carbon bonds from the precursor material. These defective structural features may subsequently be transformed into new nucleation, aggregation, and growth centres in the new interlaced carbon layers. Furthermore, the presence of some nickel nanoparticles outside the HCSs, highlighted by the small white particles, suggests the migration of Ni particles from broken HCSs or possible leaching of the Ni nanoparticles from the PSS template on the HCSs surface during the RF carbonization step.

Determination of the shell thickness of the HCS supports (with and without Ni) and the influence of the Ni-precursor salt on the size and dispersion of the Ni nanoparticles was extracted from TEM micrographs (Figure 2). The pristine HCS supports exhibited a homogeneous spherical morphology with large spheres (outer diameter (od) = 342 ± 17 nm: inner diameter (id) = 327 ± 16 nm) and a very thin carbon shell (10 ± 2.1 nm, Appendix A). On the other hand, the Ni@HCS_NiCl2_ had a thicker carbon shell (23 ± 4.7 nm) and showed the formation of large, sparsely dispersed Ni nanoparticles of ~12 ± 1.9 nm diameter (Figure 2b) on a slightly smaller HCS diameter, at (od) = 332 ± 25 nm; (id) = 281 ± 21 nm. The slightly thicker carbon shell (~23 ± 4.7 nm) after RF pyrolysis can be ascribed to the corrosion of the carbon shell on the PSS template by the precursor chloride counter anions [34]. TEM images of Ni@HCSs_Ni(acet)_ revealed that the presence of the acetate counter anions led to a HCS with an even thicker carbon shell (42.2 ± 6.9 nm), smaller nanoparticles (~6.4 ± 1.2 nm) and a HCS with similar size (od) = 326 ± 25 nm; (id) = 245 ± 19 nm) (Figure 2c, Appendix A).

It is proposed that the extra functional groups on the PAA-PSS template allowed for reaction with acetate ions and led to the thicker RF layer [35,36], and thus the thicker carbon shell. The additional functional groups also increased the number of active sites for the nucleation and growth of Ni nanoparticles resulting in the generation of smaller nanoparticles (~6.4 ± 1.2 nm, Figure 2c).

### 3.2. Textural and Thermal Stability Analysis

The textural properties of the as-synthesized pristine and Ni@HCSs samples were determined using the multi-point Brunauer–Emmett–Teller (BET) [37] method. The pore size distributions were determined using the Barrett–Joyner–Halenda (BJH) [38] method. The N_2_ adsorption/desorption isotherms (Figure 3) indicate that the samples demonstrate a type (III) isotherm with an H3 hysteresis loop at a relative pressure of P/P_0_ = 0.6–1.0. This indicates the assemblage of narrow slit-like pores or plate-like particles, suggesting the presence of both macro-and micropores within the carbon structures [12]. Based on the multi-point BET method, the specific surface areas of the samples were determined to be in the range of ~560–600 m^2^/g for the Ni@HCSs nanocatalysts, whilst the pristine HCSs samples exhibited the largest surface area at ~650 m^2^/g (Table 1).

A higher surface area was recorded in this study for the pristine HCSs as compared to hollow carbon sphere samples made using other templates such as non-porous silica [6]. This was attributed to the use of the functionalized PSS template, leading to the formation of mesopores on the shell of the pristine HCSs. Upon support of the Ni nanoparticles, the surface areas decreased, due to the blockage of the available pores on the carbon shell by the Ni nanoparticles, (Table 1). Nonetheless, the pore size distribution plots (Appendix A) illustrated a high content of pores with a diameter < 10 nm, suggesting that the available pores facilitate rapid N_2_ adsorption/desorption processes for all samples.

The thermal stability of the pristine HCSs and Ni@HCSs nanocatalysts was investigated using thermogravimetric analysis (TGA); from which it was observed that the TGA profiles were similar, (Figure 4a). For the Ni@HCSs catalysts, the complete decomposition of all samples indicates that they are all predominantly carbon based, with ~10 wt. % loading of Ni (Appendix A). To determine the decomposition profile of all the samples, first-order derivative (DTG) weight loss profiles were obtained for each sample (Figure 4b). The DTG profile of the pristine HCS exhibits a narrow decomposition peak at ~643 °C (Figure 4b, black line), indicating the uniform decomposition of the disordered nanostructures [39]. Additionally, the decomposition peak at ≤200 °C showed the presence of residual organic functional groups from the PAA-PSS template and/or RF.

Loss in thermal stability was observed for the Ni@HCSs samples in comparison to the pristine HCS samples (Appendix A), as shown by the lower onset temperatures of ~452 °C and ~426 °C for Ni@HCSs_Ni(acet)2_ and Ni@HCsS_NiCl2_ nanocatalysts, respectively. The broad DTG profiles (Figure 4b) of the Ni@HCSs were fitted to at least three decomposition peaks (Figure 4c). The decomposition component peak at lower temperatures ((i) ~484–500 °C _Ni@HCS_) is attributed to defects such as dangling bonds, structural disorder, and/or interplanar amorphous carbonaceous structures [40]. The component peak (ii) ~553–583 °C_Ni@HCS_ is associated with the decomposition of intermediate disordered carbonaceous phases, whilst (iii) the component peak at higher temperatures ~593–631 °C _Ni@HCS_) is ascribed to the decomposition of well-ordered carbon phases on the HCS support [41,42]. The thermal stability of the carbon core was found to be influenced by the presence of the anchored Ni nanoparticles, as observed by the shifted peak position as well as the slightly compromised peak ratio of the core peak with respect to that of the pristine HCS. This suggests that the Ni, in both samples, catalyses the carbon decomposition reaction.

### 3.3. Investigation of Crystallinity

The structure of the nanocatalysts synthesized using different Ni-salt precursors was determined and examined by powder XRD and Pair distribution function (PDF) analysis via total scattering measurements (Figure 5 and Figure 6). The diffraction patterns of the Ni@HCSs samples revealed both the presence of carbon structures (HCS) and Ni nanoparticles (Figure 5). This is shown by the presence of the broad diffraction peaks of polycrystalline carbon (2θ ≈ 25°) and nickel (2θ ≈ 46, 56, 77°). The diffraction peaks were indexed to the hkl (002) diffraction planes of carbon [43], as well as the (111), (200) and (220) diffraction planes of Ni [44], respectively. The Ni diffraction peak located at 2θ ≈ 46° in all samples overlapped with that of the (100) diffraction peak of polycrystalline carbon [43]; an indication of the slight ordering of the carbon lattice upon RF pyrolysis. The low degree of graphitization of the carbon shell is shown by the broadening of the peak widths relating to carbon in the diffractogram, thus illustrating the presence of amorphous carbon domains. Ultimately, the PXRD-estimated crystallite size of the Ni nanoparticles was found to be 8 and 11 nm for the Ni@HCSs nanocatalysts synthesized using Ni(acet)_2_ and NiCl_2_ precursors, respectively.

The influence of the precursor anion salt on the structure of the carbon shell was further investigated via PDF analysis of the total scattering data of the pristine HCS samples and the two Ni@HCSs nanocatalysts, as shown in Figure 6. PDF analysis is useful for distinguishing between different carbon coordination environments and is sensitive to local structure determination. Figure 6A shows a zoom of the PDF up to 11 Å, while Figure 6B,C show the PDFs up to 50 Å. As shown in Figure 6A, the signal from the HCS (black line) dampens beyond 8 Å, implying a lack of medium to long-range structural order as a result of defects and the curvature of the sheets upon the formation of nanospheres. The PDF signal from the HCS support corresponds well to that of graphene-like carbon with a density similar to graphite, as shown in the fitting curves, with the lack of a strong graphite interlayer peak at 3.64 Å. This is indicative of turbostratic disorder and misalignment of the graphene sheets along the C axis.

Carbon nearest neighbour distances of the pristine HCS sample are shown by the peaks at values of 1.42(*), 2.46(#) and 2.83(φ) Å. These peak positions correspond to the in-plane carbon–carbon bond distances in the aromatic-type ring of graphite/graphene [46,47]. For instance, the peak at 1.42(*) Å represents the sp^2^ C-C bonding of one carbon atom with its three nearest neighbours; whilst the peak at 2.46 (#) Å signifies the shortest diagonal distance between the three atoms coordinating a central carbon in the hexagonal domain of graphite or graphene. Small peak shifts to higher r values from 2.46 (#) Å show the contribution of carbonyl groups from the acetate anion, leading to an increased in-plane diagonal distance between carbon atoms. The third in-plane carbon–carbon peak at 2.83 (φ) Å signifies the long diagonal in the hexagon [48]. The broadening and shifting of these peaks increase upon the immobilization of Ni nanoparticles, thus further confirming the increased disorder in the C-ring due to the formation of defects, such as vacancies or Stone-Wales defects [49,50].

In addition to the structural modifications by the nickel anion to the carbon shell, anchoring of the nickel nanoparticles on the carbon supports revealed several structural changes in the total scattering data. Firstly, it is important to distinguish the signal originating from the Ni and that from the carbon nanostructures. The Ni–Ni bond length for FCC-Ni (2.49 Å) (#) overlaps with the distance between the three carbon atoms coordinating the central carbon or the shortest diagonal in the hexagon. Therefore, it is necessary to find a unique Ni peak where no overlap from peaks originating from the carbon nanostructures occurs. Such a peak is located at 5.55 Å (ω) and indicates that the local structure of Ni is similar in the two catalysts. In addition, the PDF signal oscillations beyond 10 Å are entirely generated from the scattering from longer range ordered Ni. This is confirmed by the PDF fits shown Figure 6B,C. It is likely that many of the Ni nanoparticles are multiply twinned as the PDF data indicates smaller crystallite sizes for both of the samples in comparison to the XRD data. The PDF fits reveal spherical particle sizes of 6.5–6.7 nm and 7.6–7.7 nm for the acetate and chloride-based samples, respectively. The decreased intensity of the intralayer C-C peak at 1.42 Å indicates increased disorder of the carbon structure upon anchoring of the Ni nanoparticles for both Ni-containing nanocatalysts.

### 3.4. Structural Analysis

The structural disorder of the HCSs originating from the anchoring of Ni nanoparticles was determined from Raman spectra. Expectedly, the spectra of all samples (Figure 7) showed first-order Raman peaks for carbonaceous materials [51,52,53], with the D-band at 1339–1350 cm^−1^ and the G-band centred at 1590–1596 cm^−1^. In addition to the common first-order Raman peaks, the spectra of the nanocatalysts also revealed the presence of other vibrational modes [54,55], as indicated by the shoulder at ~1150–1154 cm^−1^, as shown in Figure 7 by an asterisk (*). The observed shoulder (*) is assigned to sp^3^ hybridized carbons of disordered diamond-like amorphous carbon [56,57]. The position and broadness of different vibrational bands in carbon-based nanomaterials can be used as a measure for quantifying local structural disorders [58,59,60]. As such, the positions and bandwidths of all the Raman active modes within the as-synthesized nanocatalysts were determined through peak fitting of their Raman spectra. As shown in Figure 7, the spectra revealed at least four component active modes centred at ~1201–1217 cm^−1^, ~1339–1350 cm^−1^, ~1493–1511 cm^−1^, and ~1590–1596 cm^−1^, respectively. The lower frequency ~1201–1217 cm^−1^ bands result from the vibrational mode of transpolyacetylene (TPA)-like structures, indicating the formation of TPA-like chains at the zig-zag edges of defective HCS shells, with the linewidth of the TPA-band being proportional to the number of the zigzag chains [60,61,62,63]. Notably, the linewidth of the TPA-band decreased upon the addition of Ni nanoparticles to the support, and the intensity is observed to be dependent on the type of the precursor salt employed (Appendix A). For instance, the hydrocarbon radicals from the acetate anion facilitated the healing of the HCS shells (Figure 7a), leading to the formation of fewer defective zigzag edges in the Ni@HCSs_Ni(acet)2_ nanocatalysts (Figure 7b). In addition to the TPA-band, broadening of the A-band (~1493–1511 cm^−1^), especially for the Ni-based catalyst samples suggested high amorphous carbon content in the form of interstitial disordered carbon with sp^3^ links [60,63]. In particular, the etching effect of the chloride anions on the carbon shell is shown by the broader linewidth of the A-band in the Ni@HCSs_NiCl2_ nanocatalyst (Figure 7c); an indication of the incorporation of adatoms, carbon radicals, and/or sp^3^ structures into defective edges of the HCS shells.

The structural properties of the Ni@HCSs nanocatalyst were also determined through the analysis of the broadness of the D- and G-bands as well as the defect density [64,65,66,67]. Similar to the TPA-band, the linewidth of the defect-induced D-band was influenced by the interaction between the counter anion on the PAA-PSS template. For instance, the broad D-band for the Ni@HCSs_NiCl2_ nanocatalyst can be attributed to the exposure of in-plane discontinuities of the hexagonal carbon network, such as single or double vacancies, from the removal of oxygenated functional groups, absorbed radicals and/or adatoms [68,69]. On the contrary, the healing effect and possible reconstruction of the HCS shells by carbon residues from the Ni(acet)_2_ salt led to the formation of more in-plane discontinuities, thus contributing to a narrower D-band.

Finally, to determine the degree of disorder of the Ni@HCSs nanocatalysts as a function of the Ni precursor salt, the defect density ratios were determined from an estimation of the ratios of the integrated intensities of the TPA-, D- and A-band to that of the G-band (Appendix A). For all samples, the general structural defect density (I_D_/I_G_, Appendix A) ratio was ≥1, indicating a low degree of graphitization. The changes in the edge structures as a function of the Ni precursor salts were mainly highlighted by increasing I_TPA_/I_G_ and I_A_/I_G_ ratios, specifically for the Ni@HCSs_NiCl2_ nanocatalysts.

### 3.5. Surface Composition

The compositional analysis of the pristine HCSs and Ni@HCSs samples based on the XPS survey scan (Appendix A) correlated with the SEM micrographs indicating that most of the Ni nanoparticles in the Ni@HCSs_NiCl2_ sample were incorporated inside the carbon shell. The low Ni content in the Ni@HCSs_NiCl2_ nanocatalysts is attributed to the 5 nm average depth analysis of the XPS instrument, given that the carbon shell of the Ni@HCSs_NiCl2_ nanocatalysts was estimated to be ~23 ± 4.7 nm. The chemical environments of the carbon atom in the pristine HCSs and Ni@HCSs samples were determined through deconvolution of the C 1s spectra for all samples, as shown in Figure 8. The C 1s spectra were fitted to at least four component peaks corresponding to the sp^2^ C=C (284.0 eV), amorphous sp^3^ C-C (285.2 eV), defective C-OH (286.6 eV), and oxygenated carbon edges (288.9 eV) [70,71,72]. The range in the given energy values is the small shifts for each sample. To gain further insight into the correlation between the precursor salt and the Ni nanoparticles, the chemical state of nickel was determined by fitting the Ni 2p spectrum of the Ni@HCSs_Ni(acet)2_ sample (Ni 2p ~0.89 at.%, Figure 8d), despite the low counts for the Ni@HCSs_NiCl2_ nanocatalysts (0.25 at.%), the binding energy of the Ni 2p_3/2_ peak located at ~856.4–856.8 eV and that of the Ni 2p_1/2_ peaks at ~874.2–874.6 eV corresponds to the Ni^2+^ bonding states of Ni(OH)_2_ (Appendix A) [73,74,75]. Deconvolution of the O 1s spectra (Appendix A) shows the different bonding states of carbon and nickel with oxygen atoms. For instance, the pristine HCSs samples reveal the presence of vibrations corresponding to quinones (~530.1–531.1 eV), C=O and C–O bonds (~532.6–532.8 eV), as well as terminal O–H bonds (~534.1–534.3 eV) [70,71]. As ambiguous as the interpretation of the O 1s is for hybrid carbonaceous materials, the same could be said for the Ni@HCSs nanocatalysts. In particular, the peak at ~530.1 eV for the Ni@HCSs sample could also be assigned to the chemical state of Ni-O. Additionally, the peak at ~532.1 eV could be associated with the presence of O^2-^ vacancies on the nanoparticles [74,75].

### 3.6. Growth Mechanism

Generally, the results show that the nucleation of the Ni nanoparticles on the HCS shells is influenced by the surface chemistry between the PAA polymer around the PSS template and the Ni-salt precursor, as illustrated in Figure 1. For instance, the presence of carbon species in the Ni-salt precursor plays a significant role in improving the thermal stability of the Ni@HCS nanocatalysts, despite reducing their subsequent surface area. Typically, nucleation of nickel nanoparticles can be considered to follow a typical reduction of Ni^2+^ cations to metallic nickel (Equation (1)) by hydrazine in the presence of ethyl alcohol [35,36].

However, in the case of the presence of acetate counter anions, more functionalities on the PSS template via Michael addition reactions are added [76], thereby providing an abundance of active adsorption sites for nucleation, aggregation, and growth of smaller Ni nanoparticles. Furthermore, the abundance of adsorption sites also provides increased nucleation sites for carbon active species, resulting in the deposition of a thicker RF polymer and the formation of a thicker shell for the Ni(acet)_2_-based Ni/HCS catalysts.
(1)$${\mathrm{Ni}}^{2+}+N_{2}H_{4}+4{\mathrm{OH}}^{-}\to 2\mathrm{Ni}+N_{2}+4H_{2}O$$

Moreover, the extra carbon species contribute to the healing of the carbon lattice of the HCS upon heat treatment, thus leading to slightly improved thermal stability in comparison to the NiCl_2_-based nanocatalysts. Despite being a positive factor on the structural properties of Ni@HCSs_Ni(acet)2_ nanocatalysts, the additional carbon species could promote the formation of volatile nickel carbide molecules [77], which subsequently would leach upon pyrolysis and lead to the growth of Ni nanoparticles on the surface of the HCSs.

On the contrary, during the synthesis of Ni@HCSs_NiCl2_ nanocatalysts, the anchoring of Ni nanoparticles could be considered to follow a two-step mechanism. Initially, the formation of the hydrazine complex of nickel chloride is expected (Equation (2)), followed by the final reduction of the complex to a mixture of nickel hydroxide and metallic nickel nanoparticles (Equations (3) and (4)) [44,78]. As much as Ni nanoparticles are being produced, a reaction between the chloride counter anions and the dangling bonds of the PAA functional groups on the PSS is likely to occur [34]. Consequently, resulting in corrosion and/or inactivation of the carbon functionalities, thus leading to nucleation and growth of Ni nanoparticles occurring at very limited active sites and the formation of larger Ni nanoparticles.
(2)$${\mathrm{NiCl}}_{2}+{\mathrm{nN}}_{2}H_{4}↔\left[\mathrm{Ni}{\left(N_{2}H_{4}\right)}_{2}\right]{\mathrm{Cl}}_{2}$$
(3)$$\left[\mathrm{Ni}{\left(N_{2}H_{4}\right)}_{2}\right]{\mathrm{Cl}}_{2}+2{\mathrm{OH}}^{-}\to \mathrm{Ni}{\left(\mathrm{OH}\right)}_{2}+{\mathrm{nN}}_{2}H_{4}+2{\mathrm{Cl}}^{-}$$
(4)$$2\mathrm{Ni}{\left(\mathrm{OH}\right)}_{2}+N_{2}H_{4}\to 2\mathrm{Ni}+N_{2}+4H_{2}O$$

Owing to the corrosion effect of the chloride anions, the ease of decomposition for the Ni@HCSs_NiCl2_ nanocatalysts was indicated by an early on-set decomposition temperature of ~426 °C (Appendix A). However, regardless of the relatively large amount of defect sites as well as large pore diameters (~7–9 nm), the result indicated that these defects sites are not effective towards the adsorption–desorption mechanism of N_2_, leading to low surface area values for nanocatalysts based on the NiCl_2_ precursor salt. In summary, the results show the significance of the choice of the counter anion on the structural properties of the carbon-based supports for an improved design of next-generation Ni/carbon-based nanocatalysts.

## 4. Conclusions

Nickel nanoparticles were successfully anchored on the HCSs, with control of the morphologies and physiochemical properties of the HCSs supports being achieved by varying the nickel salt precursors, namely Ni(acet)_2_ and NiCl_2_. Morphological analysis with SEM and TEM revealed the influence of the counter anion on the morphology of the HCS Ni-catalyst support. For instance, the chloride anion was found to corrode the functional groups on the PSS template, thereby leading to the formation of structurally defective (I_D_/I_G_ ≈ 1.76) and thin-walled HCS (~23 ± 4.7 nm) supports. On the other hand, the acetate groups for the Ni(acet)_2_ salt precursor contributed to the functionalization of the PSS template, thus contributing more nucleation sites (I_D_/I_G_ ≈ 1.79) for the growth of small, Ni nanoparticles (~6.4 ± 1.2 nm), despite the carbon having a thick wall (~42 ± 6.9 nm). Thermal stability results suggested compromised structural integrity through the introduction of structural defects, thereby leading to easy and faster decomposition of the nanocatalysts. However, the thermal stability was found to be influenced by the contribution of the counter anion on the morphology of the carbon shell. Generally, the study shows that careful attention must be paid to the choice of the nickel metal precursor salt as this has a significant influence on the physicochemical properties of the resultant metal–carbon nanocatalysts, thus subsequently influence their catalytic activity.

## Data Availability

Not applicable.

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
