# Peer review of "Catalyst Design: Counter Anion Effect on Ni Nanocatalysts Anchored on Hollow Carbon Spheres"

_nanomaterials, 2023, doi:10.3390/nano13030426_

Round 1

Reviewer 1 Report

This paper has demonstrated a systematic study of the Ni nano-catalyst on carbon spheres. The Ni nanocrystals have been well characterized. The growing mechanism has also been studied. Both SEM and TEM have demonstrated the morphology of the catalyst. The stacking structures have also been studied by XRD and Raman spectra. The major conclusions are therefore solid. Some minor revisions must be made before the acceptance of the manuscript.

1. According to the XPS results, I agree that Ni state of Ni@HCS is Ni(II). Together with the XPS results of O, I would claim the existence of NiO. I would like to see some interpretation or discussion in the supporting information.

2. Figure 6D. Please cite VESTA if you use it for demonstrating the crystal structures. J. Appl. Crystallogr., 44, 1272-1276 (2011).

3. How about the lower angle XRD results? Does it agree with the conclusion you have for the interlayer interaction of the HCS (line 324 and line.

4. Figure 5. For pure graphite, the (200) peak is around 26.4 (2-theta). In your sample, it seems that the d-spacing is increased. Is it true? What is the implications?

5. Figure 5 blue curve. The C/Ni overlapped peak could be C-(101) peak instead of (100). Please check the standard PDF card. Both C-(100) and C-(101) should have overlapped with Ni-(111). (if it is graphite-like and Ni is FCC symmetry).

Anyway, the rest of the results can be published without revision. This article does provide some new insights into the preparation of Ni-nanoparticles on carbon-based substrates.

Author Response

Kindly check the attachment.

Reviewer 2 Report

The investigation made by the authors is interesting, enough comprehensive and important for the fabrication of Ni@NHCs. The results are presented clearly and the goals of the study are achived. There are some points which should be addressed by the authors prior to publication of this manuscript:

1) Ni(OAc)2 would be much better instead of Ni(acet)2/

2) The indication that the adsorption of N2 is studied should be added somewhere on Figure 3.

3) Why are not the molecules of hydrazine depicted on Scheme 1 in the case of Ni(OAC)2 precursor (unlike in the case with NiCl2 precursor)? They can be shown as separate molecules, without forming the complexes with Ni(II). Nevertheless, it is reasonable to suppose that Ni(II) cations partially form the chelates (shown on Scheme 1) and partially form complexes with hydrazine.

4) 3.5. Surface Composition. Information is not very clear about the presence of oxygen in the materials: C-O, C=O and Ni-O bonds are mentioned but the amount of O atoms, possible presence of NiO is not discussed at all.

5) 10% Wt. content of Ni is postulated in the beginning of the Experimental Part. As no elemental analysis data are provided, the authors derive the weight content of Ni from the thermal stability properties (Table S2). Does it mean that Ni@HCSs contain ca 6% wt. of Ni in the case of NiCl2 precursor and ca 10% wt. in the case of Ni(OAc)2. This information is very important and should be given more explicitly, in the Discussion as well in Conclusions sections.

6) It is evident that the authors did not investigate catalytic performance of the obtained Ni@HCSs, however it would be helpful for the readers to outline possible areas of application of such nanocatalysts (e.g. in Conclusions section).

Author Response

Attachment

Reviewer 3 Report

The paper “Catalyst Design: Counter anion effect on Ni nanocatalysts anchored on hollow carbon spheres” is interesting, on a high-interest topic, well written and explained. I recommend that the paper be accepted after minor revision. Some recommendations and comments are made below:

Keywords: “ccatalyst design; ounter anion” – needs correction;

Introduction:

- rows 42-42: “many carbon allotropes have been explored as popular candidates in electrocatalysis as metal catalyst supports” – please give some examples and references;

2. Materials and Methods. 2.2. Synthesis of Ni@HCSs nanocatalyst

- row 125: the authors use PSS (then many more times in the manuscript) without explaining it; it probably means polystyrene spheres, but it sould be explained;

- a short explanation for the reason of choosing nickel acetate and chloride as nickel precursors would be beneficial for the reader. Usually in catalysis nitrates or organic salts (such as acetates, oxalates etc.) are chosen, but rarely chlorides because, if they are not completely removed during washing, by calcination the chloride anion is not removed from the final solid, contrary to other anions that decompose into gas molecules and do not pose problems during the catalytic process itself;

3. Results and Discussion.3.1 Morphological Analysis

- rows 197-199: “On the other hand, a few defective spheres were also evident as seen by the presence of ‘cup-like’ structures is (i) an indication of incomplete carbonization of the polystyrene template spheres (PSS) with the resorcinol formaldehyde …” – why would incomplete carbonization lead to defective spheres?

- rows 225-226: “outer diameter (od) = 342 ± 17 nm: inner diameter (id) =327 ± 16 nm) and a very thin carbon shell (10 ± 2.1 nm)” and the following few rows explaining Figure 2: usually we expect that the difference between the outer dimeter and inner diameter to be the thickness of the shell, but the reader gets confused by the numbers, that don’t add up; please give a short explanation;

3.2 Textural and thermal stability analysis

- row 275: “narrow decomposition peak at ~568 °C” – the black line peak in Figure 4(b) is clearly above 600 °C, please correct;

3.4 Structural Analysis

- rows 371 and 377: the authors discuss Figure 7, but write “figure 5”; correction necessary;

3.6 Growth Mechanism

- rows 473-474: “Initially, the formation of the hydrazine complex of nickel chloride is evitable …” – what do authors mean by that?
